# Secure Split Learning against Property Inference and Data Reconstruction Attacks

## Abstract

Split learning of deep neural networks (SplitNN) has provided a promising solution to learning jointly for the mutual interest of a guest and a host, which may come from different backgrounds, holding features partitioned vertically. However, SplitNN creates a new attack surface for the adversarial participant, holding back its practical use in the real world. By investigating the adversarial effects of two highly threatening attacks, i.e., property inference and data reconstruction, adapted from security studies of federated learning, we identify the underlying vulnerability of SplitNN. To prevent potential threats and ensure learning guarantees of SplitNN, we design a privacy-preserving tunnel for information exchange between the guest and the host. The intuition behind our design is to perturb the propagation of knowledge in each direction with a controllable unified solution. To this end, we propose a new activation function named $R^3$eLU, transferring private smashed data and partial loss into randomized responses in forward and backward propagations, respectively. Moreover, we give the first attempt to achieve a fine-grained privacy budget allocation scheme for SplitNN. The analysis of privacy loss proves that our privacy-preserving SplitNN solution provides a tight privacy budget, while the experimental result shows that our solution outperforms existing solutions in attack defense and model usability.

## 1 Introduction

Collaborative learning enables participants from different backgrounds to learn jointly for mutual interests. A well-known collaborative learning paradigm is federated learning (FL) [25], focusing on the coordination of distributed participants. Meanwhile, another paradigm, split neural network learning (SplitNN for short) [19, 11, 4], is designed for vertically partitioned features. The emergence of SplitNN provides a promising solution to building cooperative models such as two-towers recommendation systems [37, 39]. By combining different features, SplitNN is supposed to be more expressive for tasks like user profiling and recommendation.

However, collaborative learning paradigms are faced with severe security issues. Roughly speaking, there are three kinds of known threats, inference attack [9, 30], reconstruction attack [17, 31], and poisoning attack [33, 18]. The inference attack discloses properties or membership information of data samples, while the reconstruction attack seeks to recover participants' private data samples. Unlike these two kinds of threats, the poisoning attack [33, 18] aims to put harmful data into collaborative learning for malicious purposes rather than stealing private information. As a result, some defensive solutions have been proposed for federated learning. According to their techniques, these solutions can be classified into three categories: differential privacy solutions [36, 32], homomorphic encryption solutions [38, 21], and secure multiparty computation solutions [12, 27].

Submitted to 36th Conference on Neural Information Processing Systems (NeurIPS 2022). Do not distribute.

Unfortunately, defense solutions for SplitNN are barely discussed, but threatening attacks are continuously emerging. We note that the workflow of SplitNN has a unique asymmetric design, which is quite different from FL. Therefore, most solutions for secure FL are not suitable for SplitNN. Secure multiparty computation solutions and homomorphic encryption solutions can achieve ideal data confidentiality in SplitNN. But the overhead introduced is still far away from practical uses. Thus, for the first attempt at a secure SplitNN solution, we will concentrate on privacy leakage issues caused by property inference and data reconstruction attacks in this paper because they share similar adversarial goals. Instead, poisoning attacks need to be studied separately [7, 24].

We note an inherent contradiction [6] between privacy preservation and model utility in SplitNN. To investigate the privacy leakage risk of SplitNN, we evaluate the adversarial effect of property inference and data reconstruction attacks against recommendation and image classification models built using SplitNN. The results in semi-honest and malicious settings show that both attacks have sufficiently high success rates when disclosing the privacy of the guest or the host. To secure SplitNN against these attacks [17, 30, 29], we give the first privacy-preserving SplitNN solution, stemming privacy leakage from either direction with dynamic privacy budget allocation.

**Contribution**. Through a thorough investigation of the privacy leakage issue in SplitNN, we confirm that the best option of defensive solution for SplitNN is to construct a privacy-preserving tunnel between the split surfaces of the host and guest sides. To this end, we propose a novel activation function named $R^3eLU$, responding to forward and backward propagations in a randomized manner. Furthermore, we propose a fine-grained privacy budget allocation scheme for SplitNN to achieve more efficient perturbations, dynamically allocating the privacy budget in spatial dimension (feature level) and temporal dimension (epoch level). The analysis of privacy loss shows that our solution provides a differential privacy guarantee regarding activations. The evaluation results regarding recommendation and classification models show that our solution can outperform the existing solutions in privacy preservation and model usability.

# 2 Problem Statement

## 2.1 Split Learning

Given a training dataset $X$ and model parameters $\theta$, a learning task is to find approximately optimal $\theta$ by minimizing a pre-defined loss function $\mathcal{L}$. We assume that the optimizer used is a mini-batch stochastic gradient descent (SGD) algorithm, which updates $\theta$ with a batch input of $X$ iteratively. Assuming the batch size is $N$, then the total loss of $\theta$ for a batch input $x = \{x_i | x_i \in X, i \in [1, N]\}$ should be $\sum_{x \in x} \mathcal{L}(\theta, x)$ in the $t$-th training iteration. The gradients of $\theta$ for model updating should be estimated by $\frac{1}{N} \sum_{x \in x} \nabla_{\theta} \mathcal{L}(\theta, x)$ approximately. Hence, parameters $\theta$ can be updated as $\theta^{t+1} = \theta^t - \frac{1}{N} \sum_{x \in x} \nabla_{\theta} \mathcal{L}(\theta, x)$. This mini-batch SGD based optimizing procedure should be repeated until the model usability meets the requirement or the maximal count of iterations reaches.

Generally, there are two roles in a two-party SplitNN. We denote by the guest who holds features only and the host who holds both features and labels. According to related studies [34, 14], there exist several split configurations of neural networks. We will focus on the SplitNN designed for vertically partitioned features with labels held by the host only [4, 11]. We denote by $\theta^g$, $\theta^h$ the partial models of the guest and the host after split and $\theta^k$ the rest part of the original model. Then the forwarding result $\texttt{Forward}(x^g, \theta^g)$ of the guest should be evaluated locally and passed to the host. The host should merge $\texttt{Forward}(x^h, \theta^h)$ with $\texttt{Forward}(x^g, \theta^g)$ using a predefined strategy such as concatenating and averaging. Then the host finishes the rest of the forward propagation $\mathcal{L}(\texttt{Merge}(\texttt{Forward}(x^g, \theta^g), \texttt{Forward}(x^h, \theta^h)), \theta^k)$ and initiates backward propagation, sending the partial loss regarding $x^g$ and $\theta^g$ back to the guest.

We take a SplitNN based recommendation system as an example and give a benchmark in Table 1. Different merging strategies and two public datasets, MovieLens [15] and BookCrossing [40], have been evaluated. In case of misaligned features, zero padding will be used to retain the shape of fea-

ture vectors. We notice that concatenating and element-wise averaging with padding have relatively stable and desirable performance, which will be used as the default setting in the rest of this work.

Table 1: Top-10 hit ratio of SplitNN based recommendation using different merging strategies.

| | | concatenate | element-wise | | | | no split |
| | | | max | sum | average | min | |
|---|---|---|---|---|---|---|---|
| MovieLens | padding | 56.62% | 56.26% | 56.35% | 56.89% | 57.19% | 57.21% |
| | non-padding | 55.38% | 54.75% | 54.95% | 55.72% | 55.08% | |
| Book Crossing | padding | 61.70% | 60.84% | 60.21% | 61.16% | 60.98% | 61.92% |
| | non-padding | 58.80% | 59.34% | 59.44% | 59.10% | 59.02% | |

## 2.2 Threat Model

In SplitNN, interactions between the guest and the host pose threats to each other. Thus, we will investigate private data leakage threats from either direction. We first assume that the host and the guest are honest but curious about the private data of each other. Moreover, we will consider a more powerful threat model [29], where the guest or the host could be malicious, hijacking the feature space during split learning. In both cases, we take into account property inference and data reconstruction attacks, which are highly threatening attacks identified in collaborative learning.

*Property inference attack*. Since either role of SplitNN has access to the output of the other's local model, the adversary can mount a property inference attack [9, 22], inferring properties of private data through observing query input and the corresponding output. By constructing shadow models elaborately, the adversary can steal substantial information from the target. In this way, the adversary acquires the capability of inferring some properties (such as gender and age) of the data samples used for training. Denoted by $F$, $T$ and $\mathcal{L}_F$ the inference model, target model, and the loss function used for $F$, the adversarial goal of property inference attack is

$$\mathcal{A}_{PIA} = \arg\min_F \sum_{x_i \in \boldsymbol{x}} \mathcal{L}_F(F(T(x_i)), l_i), l_i \in \{0, 1\}. \tag{1}$$

*Data reconstruction attack*. By taking advantage of generative adversarial networks (GANs) [13], a data reconstruction attack [17, 29] becomes possible in collaborative learning. To mount the attack, the adversary augments the training data per iteration by inserting fake samples $\boldsymbol{z}$ generated by a generator $G$. The target model will serve as a discriminator $D$. The adversary affects the target model by deceiving the target with fake training samples. For correcting the adversary, the target participant is supposed to put more private information into the learning. In this game-style training, the adversary can obtain a generator to reconstruct data samples similar to target private data. The adversarial goal of the data reconstruction attack can be given as

$$\mathcal{A}_{DRA} = \min_G \max_D \frac{1}{|x|} \sum_{x \in \boldsymbol{x}} log D(x) + \frac{1}{|\boldsymbol{x}|} \sum_{z \in \boldsymbol{z}} log(1 - D(G(z))). \tag{2}$$

*Feature space hijacking*. The property inference and data reconstruction attacks adapted from FL can be mounted by a semi-honest participant, who follows the split learning protocol normally. However, a recent attack study dedicated to SplitNN has revealed that a malicious host can achieve more impressive adversarial effects on property inference or data reconstruction by explicitly distorting the objective of split learning. Thus, we also consider the defensive effect of our solution against property inference and data reconstruction attacks using this feature space hijacking approach.

## 3 Privacy-Preserving Split Learning

Our goal is to design an unified defense solution to preserving privacy from both the host and guest's perspectives. Ideally, the guest wants to collaborate with the host under the condition that the host could disclose no private information and vice versa. According to recent studies [3, 23, 16], artificial perturbations of data samples or parameters could prevent privacy leakage effectively. However,

different from conventional model publishing scenarios, the host and guest in SplitNN will keep exchanging intermediate results during training. These continuous queries significantly increase the risk of privacy leakage for both sides. Moreover, the attack surface of splitNN is in the middle of neural network propagations, which makes things tricky. Thus, our primary idea to tackle this problem is to construct a bidirectional privacy-preserving tunnel for interactions. Recent studies such as [10] have proved that activation functions are more adaptive for perturbed operands than other neural network components. Moreover, activation functions have various forms [2], which are flexible for configuration. As a result, we propose a new variant of ReLU as a privacy-preserving interface.

## 3.1 $R^3$eLU: Randomized-Response ReLU

Inspired by a randomized response approach [35], we propose a new activation function named $R^3$eLU. The original randomized response method is good at statistical analysis of item sets. But the result of an activation function is commonly a continuous variable. Hence, they cannot be easily combined together. It should also be noted that it is risky to perturb activation functions directly because non-activated results may be flipped unexpectedly. Recall that the original formula of ReLU is $f(x) = \max(0, x)$, $x \in \mathcal{R}$. Our randomized-response variant will yield a proper substitute for replacing the real activation with a probability of $p$. If we yield $0$ as the substitute, then we can randomly inactivate a part of ReLU results. But nothing has been changed for $x \leq 0$. Because $f(x) = 0$ when $x \leq 0$. Hence, the variant is not completely privacy preserved since $f(x) = 0$ also reveals useful information to the adversary. For the completeness of the variant, we generate noisy activations $x' \leftarrow \text{Laplace}(0, \sigma)$ for $x \leq 0$. Now, we can give the definition of $R^3$eLU as

$$R^3eLU(x) = \begin{cases} \max(0, x + x'), & \text{with probability } p, \\ 0, & \text{with probability } (1-p). \end{cases} \tag{3}$$

We remark that the way $R^3$eLU handles non-activated results is dangerous, although learning accuracy is often traded off for privacy. But we will mitigate the side effect through privacy budget allocation. Then the risk of applying $R^3$eLU will not be a problem.

## 3.2 Forward Propagation with $R^3$eLU

Now we show how to apply $R^3$eLU in the forward pass of SplitNN regarding the guest's privacy. Generally, when the $t$-th training iteration begins, the guest randomly samples a mini-batch $\boldsymbol{x}$ from private training dataset $\mathcal{X}^{(g)}$. Assuming that an embedding procedure $\texttt{Embedding}() : \mathcal{R}^M \leftarrow \mathcal{X}$ is publicly available, raw data samples in the mini-batch can be encoded into feature vectors $V = \{\boldsymbol{v}_1, \boldsymbol{v}_2, \ldots, \boldsymbol{v}_N\}$, $\boldsymbol{v}_i = \{v_1, v_2, \ldots, v_M\}$, $i \in [1, N]$, and $M$ is the dimension of feature representing space. From the functional perspective, we assume that $\mathcal{L}(\boldsymbol{v}^h, \boldsymbol{v}^g, \boldsymbol{\theta})$ is equivalent to $\mathcal{L}(\texttt{Forward}(\boldsymbol{v}^h, \boldsymbol{\theta}^h), \texttt{Forward}(\boldsymbol{v}^g, \boldsymbol{\theta}^g), \boldsymbol{\theta}^k)$, where $\texttt{Forward}() : \mathcal{R}^{N_s} \leftarrow \mathcal{R}^M$ is a function to yield the feedforward result of hidden units for a given neural network, and $N_s$ indicates the output shape.

We now make the minimal modification of the forward propagation of SplitNN while leaving the rest part unchanged. Denoted by $\texttt{Forward-R}^3\texttt{eLU}()$ the feedforward result of neural network by replacing the activation functions of the guest's output layer with $R^3$eLU. Then the feedforward result transmitted to the host should be $\boldsymbol{a}^g = \texttt{Forward-R}^3\texttt{eLU}(\boldsymbol{v}^g, \boldsymbol{\theta}^g)$. Next, the host executes a predefined aggregation procedure taking as input $\boldsymbol{a}^g$ and $\boldsymbol{a}^h$. Finally, the loss function will be evaluated by $\mathcal{L}(\texttt{Forward-R}^3\texttt{eLU}(\boldsymbol{v}^g, \boldsymbol{\theta}^g), \texttt{Forward}(\boldsymbol{v}^h, \boldsymbol{\theta}^h), \boldsymbol{\theta}^k)$. Algorithm 1 in the appendix integrates the above forward propagation using $R^3$eLU into SplitNN.

## 3.3 Privacy-Preserving Backward Propagation

According to recent studies of privacy leakage in backward propagation [26, 31, 8], model updating information may cause severe leakage of private training data. Since the partial loss will be propagated to the guest in SplitNN, it is crucial to protect the host's privacy from being disclosed. Fortunately, our $R^3$eLU is adaptable to noisy partial losses and we design $R^3$eLU-Diff in a randomized-

response manner for the derivative of R³eLU as

$$\text{R}^3\text{eLU-Diff}(\boldsymbol{\delta}^g, \boldsymbol{a}^g, \boldsymbol{v}^g) = \begin{cases} \boldsymbol{\delta}^g \times \text{ReLU-Diff}(\boldsymbol{a}^g, \boldsymbol{v}^g) + x', & \text{with probability } (1-p), \\ 0, & \text{with probability } p. \end{cases}$$

Noting that the derivative value of ReLU for any input is either one or zero, randomly flipping the derivative value may still disclose $\boldsymbol{\delta}_t^g$ when value ones are not flipped. Therefore, we integrate a Laplace mechanism into the R³eLU-Diff. Please also note that the constructions of R³eLU-Diff and R³eLU are in a similar way. This is helpful to obtain uniform analysis results of two parties, which will be shown in the privacy analysis part.

We sketch the backward propagation using R³eLU-Diff in Algorithm 2 in the appendix. Briefly, when the $t$-th backward propagation begins, the host calculates the partial loss $\boldsymbol{\delta}^k = \nabla_{\boldsymbol{a}^k} \mathcal{L}(l, \bar{\boldsymbol{a}}, \boldsymbol{\theta}_t^k)$ for $\boldsymbol{\theta}^k$ regarding the total loss $l$ obtained in forward propagation, where $\bar{\boldsymbol{a}}$ is the averaged activation result. Then a $\texttt{Backward}() : \mathcal{R}^{N_r} \leftarrow \mathcal{R}^{N_s}$ procedure calculates the gradients of model parameters, where $N_r$ is the shape of parameters. To update the guest model, the host sends partial loss $\boldsymbol{\delta}^g$ for $\boldsymbol{\theta}^g$ to the guest. To preserve the host's privacy within $\boldsymbol{\delta}^g$, we disturb the partial loss $\boldsymbol{\delta}^g$ propagating to the guest and keep the partial loss on the host side unchanged.

### 3.4 Dynamic Privacy Budget Allocation

It has been proved that the importance of a parameter can be quantified by the error introduced when it is removed from the model [28]. Thus, we define the importance $I_j$ of a given SplitNN parameter $\theta_j \in \boldsymbol{\theta}$ as the squared difference of prediction errors caused by removing $\theta_j$,

$$I_j = (\mathcal{L}(\boldsymbol{x}, \boldsymbol{\theta}) - \mathcal{L}(\boldsymbol{x}, \boldsymbol{\theta} \setminus \{\theta_j\}))^2. \tag{4}$$

Due to the consideration of efficiency, it is suggested in [28] to estimating the importance by a first-order Taylor expansion approximately. Then the importance of $\theta_j$ is estimated as

$$\hat{I}_j = (g_j \cdot \theta_j)^2, \tag{5}$$

where $g_j$ is the gradient of the parameter $\theta_j$ regarding a specific sample when $\boldsymbol{\theta}$ is well-trained. Given parameter importance, the importance of features can be derived further. Specifically, the importance of a neuron (or a feature) $U_j$, $j \in [1, N_u]$, where $N_u$ is the total number of neurons in the model, can be calculated as a joint importance of relevant parameters by summing up the importance of all relevant parameters. Hence, $U_j = \sum_{\theta_k \in \tilde{\boldsymbol{\theta}}_j} \hat{I}_k$, where $\tilde{\boldsymbol{\theta}}_j$ denotes the set of all parameters directly connected to the $j$-th neuron.

However, the above importance estimation method is designed for a well-trained model and cannot be directly applied to intermediate models during training. To tackle this problem, we give a dynamic estimation method by deriving the original method into a cumulative form. The importance of a feature will be accumulated as the training epoch increases. Specifically, the importance of the $j$-th feature in the $q$-th training epoch is

$$U_j^q = \frac{\sum_{\theta_k \in \tilde{\boldsymbol{\theta}}_j} \hat{I}_k + U_j^{q-1} \times (q \times \lfloor T/n_t \rfloor + (t \bmod n_t) - 1)}{q \times \lfloor T/n_t \rfloor + (t \bmod n_t)}, \tag{6}$$

where $n_t$ indicates the iteration number within a training epoch, $T$ is the maximum training iteration number, and $t$ is the current training iteration globally. Generally, we assume that $T/n_t = n_q$, $n_q \in \mathcal{N}$, which also means that $q \in [1, n_q]$. We give the importance estimation results of different neurons in Figure 3 in the appendix, showing the correctness of our dynamic importance estimation method and the existence of unbalanced feature importance.

Given the parameter importance estimated dynamically, we are capable of allocating privacy budgets regarding different features. The intuition is to give larger budgets to more important features while smaller budgets to less important ones. Before the $q$-th training epoch begins, we estimate the feature importance vector $\boldsymbol{U} = \{U_1^q, U_2^q, \ldots, U_{N_u}^q\}$. Based on $\boldsymbol{U}$, the corresponding privacy budget to be allocated should be $\epsilon_j \times U_j^q$, $j \in [1, N_u]$. And the total privacy budget for all features is

$\epsilon_F = \sum_{j=\in[1,N_u]} \epsilon_j$. On the other hand, we can also dynamically allocate privacy budgets for different iterations to optimize the total privacy budget further. Given the total privacy budget $\epsilon_T$ for all iterations, we assign the privacy budget $\epsilon_i = \frac{\epsilon_T}{2^i}$ to the $i$-th iteration. Since $\sum_{i=1}^{\infty} \frac{\epsilon_T}{2^i} = \epsilon_T$, according to the sequential composition theory of differential privacy, we can still ensure that the whole training process achieves $\epsilon_T$-differential privacy.

## 3.5 Privacy Analysis

We will give the privacy analysis for the host and the guest respectively. In forward propagation, we recall that $\boldsymbol{v}^g$, $\boldsymbol{\theta}^g$, and $\boldsymbol{a}^g$ are input features, model parameters, and activation results of the guest. According to the definition of R$^3$eLU, $\boldsymbol{a}^g$ should be randomly flipped with probability $p$. For brevity, we denote by $f()$ the evaluation of neural network before activation. Then the probability of observing any activation $a_o$ for a given input $v$ should be

$$P(a_o = 0|v) = p + (1-p) \int_{-\infty}^{-f(v)_o} \frac{1}{2b} \exp(-\frac{|v|}{b}) dv = p + \frac{1-p}{2} \exp(\frac{-f(v)_o}{b}),$$

$$P(a_o > 0|v) = \frac{1-p}{2b} \exp(-\frac{|a_0 - f(v)_o|}{b}).$$

Then we can give the following conclusion regarding the guest model. The proof is in the appendix.

**Corollary 1.** *When forward propagation of the guest model is activated by R$^3$eLU in split learning, the activation result is $\epsilon$-DP, given Laplace noise scale $\sigma_g$.*

For a fine-grained privacy budget allocation, we introduce dynamic budget allocation in our solution. We now give the analysis of feature-specific privacy budget. Given the estimated feature importance vector $U = \{U_1^q, U_2^q, \ldots, U_{N_u}^q\}$, we allocate privacy budget $\epsilon_i = \epsilon U_i^q$ to each feature. Thus, if the divergence of features can be bounded by their privacy budgets, then the total privacy budget will be bounded. Denoted by $b_i$ and $c_i$ the noise parameter and bound of the $i$-th feature. Then we can account the divergence of features as

$$\frac{P(a_i > 0|v)}{P(a_i > 0|v')} = \frac{\frac{1-p}{2b} \exp(-\frac{|a_i - f(v)_i|}{b_i})}{\frac{1-p}{2b} \exp(-\frac{|a_i - f(v')_i|}{b_i})} \leq \exp(\frac{|a_i - f(v')_i| - |a_i - f(v)_i|}{b_i}) \leq \exp(\frac{2c_i}{b_i})$$

$$\frac{P(a_i = 0|v)}{P(a_i = 0|v')} = \frac{p + \frac{1-p}{2} exp(\frac{-f(v)_i}{b_i})}{p + \frac{1-p}{2} exp(\frac{-f(v')_i}{b_i})} \leq \frac{p + \frac{1-p}{2} exp(\frac{c_i}{b_i})}{p + \frac{1-p}{2} exp(\frac{-c_i}{b_i})} \leq \exp(\frac{2c_i}{b_i})$$

Basing on this result, we can conclude the following corollary.

**Corollary 2.** *In forward propagation with R$^3$eLU, the privacy budget of the $i$-th feature can be bounded by $\epsilon_i$ if we choose $b_i$ to satisfy $\exp(\frac{2c_i}{b_i}) \leq \exp(\epsilon_i)$, $\forall p \in [0, 1]$.*

The last analysis result of the guest is privacy budget allocation during the whole training stage. Since we have allocated $\epsilon = \sum_{i\in[1,N_u]} \epsilon_i$ for all features, we can directly conclude that each training step is $\gamma\epsilon$-DP by following the privacy amplification theory, where $\gamma = \frac{B}{N}$ is the sampling ratio of a batch regarding the whole training dataset. Now we can give the total privacy budget of the whole training stage using the strong composition theorem.

**Corollary 3.** *The total privacy budget of the whole training process using R$^3$eLU is $(\epsilon_g', \delta_g')$-DP, where $\epsilon_g' = \gamma\epsilon\sqrt{2T \ln(\frac{1}{\delta_g'})} + \gamma\epsilon T(e^{\gamma\epsilon} - 1)$.*

So far, we have given the privacy analysis from the guest perspective. Since we construct R$^3$eLU and R$^3$eLU-Diff using the same way and same technique, these two procedures have the same analysis result if we choose the noise scale of host $\sigma_h = \sigma_g$. In this way, we can conclude the following result for the host.

**Corollary 4.** *In backward propagation with R$^3$eLU-Diff, the privacy budget of the host can be bounded by $\epsilon_h$ if we choose $\sigma_h = \sigma_g$. The total privacy budget of the host in the whole training process is $(\epsilon_h', \delta_h')$-DP, where $\epsilon_h' = \gamma\epsilon_h\sqrt{2T \ln(\frac{1}{\delta_h})} + \gamma\epsilon_h T(e^{\gamma\epsilon_h} - 1)$.*

## 4 Evaluation

We evaluate our privacy-preserving SplitNN solution from two aspects model usability and privacy loss. To be comprehensive, we will compare our solution with the baseline (without any protection) and the most relevant defense solutions, i.e., a primitive Laplace mechanism [5] and DPSGD [1], the most well-known privacy-preserving deep learning solution, in the same setting. We will use the same fixed total privacy budget and the same split way (shown in the appendix) for all solutions. We adapt solutions into recommendation models using two real-world datasets, MovieLens [15], BookCrossing [40], and an image classification model using MNIST [20] dataset. The MovieLens 1-M dataset contains 1 million ratings of 4,000 movies collected from 6,000 users and users' demographic information such as gender and age. The BookCrossing dataset includes 278,858 users' demographic information and 1,149,780 ratings of 271,379 books. The MNIST database has 70,000 handwriting image examples. We will use a 32 batch size, a 0.01 learning rate and an Adam optimizer as default. Since different datasets and defense solutions may require various epochs for split learning, we calculate the metrics when the learning converges, or the privacy budget is drained. All experimental results are averaged across multiple runs.

Table 2: Model usability results while preserving the privacy of the guest.

| $\epsilon$ | MovieLens | | | BookCrossing | | | MNIST | | |
|---|---|---|---|---|---|---|---|---|---|
| | Laplace | DPSGD | **Ours** | Laplace | DPSGD | **Ours** | Laplace | DPSGD | **Ours** |
| 0.1 | 30.84% | 32.29% | **34.03%** | 57.02% | 55.89% | **58.18%** | 17.43% | 30.21% | **32.41%** |
| 0.5 | 41.25% | 43.69% | **43.87%** | 57.67% | 56.14% | **58.54%** | 27.33% | 58.43% | **60.38%** |
| 1.0 | 48.16% | 49.09% | **50.56%** | 58.02% | 56.56% | **58.42%** | 31.05% | 75.58% | **76.60%** |
| 2.0 | 49.32% | 50.38% | **50.49%** | 58.74% | 56.91% | **59.24%** | 38.92% | 92.90% | **93.53%** |
| 4.0 | 49.26% | **50.86%** | 50.73% | 59.01% | 57.16% | **59.26%** | 95.37% | **95.87%** | 94.12% |

Table 3: Model usability results while preserving the privacy of the host.

| $\epsilon$ | MovieLens | | | BookCrossing | | | MNIST | | |
|---|---|---|---|---|---|---|---|---|---|
| | Laplace | DPSGD | **Ours** | Laplace | DPSGD | **Ours** | Laplace | DPSGD | **Ours** |
| 0.1 | 31.47% | 30.68% | **33.98%** | 57.37% | 57.46% | **58.26%** | 27.64% | **33.45%** | 32.36% |
| 0.5 | 41.75% | 42.31% | **42.67%** | 58.62% | 58.24% | **58.59%** | 55.38% | 65.28% | **67.83%** |
| 1.0 | 47.43% | 48.29% | **50.39%** | 59.49% | 58.44% | **59.77%** | 71.95% | **89.74%** | 88.14% |
| 2.0 | 49.86% | 50.43% | **51.47%** | 59.34% | 59.97% | **60.27%** | 89.15% | **92.66%** | 92.52% |
| 4.0 | 49.57% | 50.09% | **51.62%** | 59.55% | **60.75%** | 60.66% | 94.61% | **95.37%** | 95.01% |

### 4.1 Model Usability

Since artificial perturbation may affect the learning procedure, we will evaluate how SplitNN is affected by privacy-preserving solutions. Moreover, two asymmetric parties of SplitNN may have different privacy concerns and affect learning differently. Thus, we will evaluate model usability concerning privacy from the perspective of the guest or the host. We use an averaged test accuracy across all test samples for the evaluation. Precisely, the test accuracy of a recommendation model is calculated using a top-10 hit ratio, while the test accuracy of an image classifier is the prediction accuracy. In Table 2 and Table 3, we show the model usability results regarding various privacy budget values of two parties. We note that model accuracy baselines of Movielens, BookCrossing, and MNIST are 56.62%, 61.70%, and 98.00%, respectively.

For the MovieLens model, our solution achieves the best model usability in most cases, especially with less privacy budget. DPSGD has a better result when $\epsilon = 4$ for the guest. But DPSGD will cause a significant privacy leakage in this case. For the BookCrossing model, the model usability of our solution is relatively high in cases of protecting the guest and the host. Similarly, DPSGD has a better result when $\epsilon = 4$ for the host, sacrificing the privacy guarantee. Things are a bit different for the MNIST model. DPSGD gets some better results when protecting the host. The reason is that split learning for an image classification model segments image samples roughly, making our dynamic budget allocation approach malfunction. Meanwhile, DPSGD is not designed for protecting partial loss in SplitNN, leading to an optimistic estimation of the threat against the host. Apart from these exceptional cases, our solution outperforms other solutions on model usability.

## 4.2 Defense Against Inference and Reconstruction Attacks

We will evaluate the performance of privacy preservation by comparing attack results against SplitNN with and without the defense. We will mount property inference and data reconstruction attacks against the guest and the host, respectively. Prediction accuracy of the adversary's inference model will be used to measure the performance of the property inference attack. As for the data reconstruction attack, the adversary tries to generate data samples as similar as possible to the target's private data. In this case, we can use a mean squared error (MSE) between a generated sample and a target data sample to measure the adversarial effect.

Table 4: Results of defending the guest against property inference attack.

| $\epsilon$ | MovieLens | | | BookCrossing | | | MNIST | | |
|---|---|---|---|---|---|---|---|---|---|
| | Laplace | DPSGD | **Ours** | Laplace | DPSGD | **Ours** | Laplace | DPSGD | **Ours** |
| 0.1 | 66.99% | 77.71% | **60.99%** | **54.76%** | 73.29% | 55.78% | **43.27%** | 53.95% | 44.33% |
| 0.5 | 66.16% | 74.23% | **64.16%** | **54.97%** | 74.52% | 56.33% | 46.92% | 54.23% | **45.59%** |
| 1.0 | **67.19%** | 78.65% | 68.18% | **55.03%** | 74.96% | 58.65% | 47.58% | 54.26% | **47.51%** |
| 2.0 | 68.65% | 73.06% | **68.56%** | **54.85%** | 74.26% | 58.14% | **48.06%** | 54.65% | 52.87% |
| 4.0 | **69.14%** | 76.18% | 71.91% | **54.92%** | 74.33% | 60.76% | **48.47%** | 54.57% | 55.73% |

Table 5: Results of defending the host against property inference attack.

| $\epsilon$ | MovieLens | | | BookCrossing | | | MNIST | | |
|---|---|---|---|---|---|---|---|---|---|
| | Laplace | DPSGD | **Ours** | Laplace | DPSGD | **Ours** | Laplace | DPSGD | **Ours** |
| 0.1 | 53.46% | 78.59% | **51.86%** | **54.55%** | 74.35% | 59.42% | 60.34% | 80.29% | **48.74%** |
| 0.5 | 53.46% | 75.64% | **51.89%** | **54.62%** | 74.36% | 59.42% | 59.82% | 81.92% | **49.71%** |
| 1.0 | 53.46% | 73.54% | **52.75%** | **54.95%** | 74.39% | 59.52% | 59.74% | 82.80% | **50.48%** |
| 2.0 | **53.47%** | 75.05% | 59.77% | **54.40%** | 74.39% | 58.13% | 60.38% | 88.88% | **50.57%** |
| 4.0 | **53.48%** | 79.28% | 56.52% | **54.95%** | 74.39& | 62.04% | 60.62% | 89.73% | **50.47%** |

***Defense against property inference attack***. We give evaluation results of the defensive effect of the guest and the host in Table 4 and Table 5, respectively, inferring the property of users in MovieLens and BookCrossing and an image patch in MNIST. The attack accuracy against baselines of MovieLens, BookCrossing, and MNIST models can achieve above 80%, 79%, and 94% by an adversarial host, 80%, 78%, and 57% by an adversarial guest, respectively. However, our SplitNN solution can effectively mitigate the adversarial effect during training and decrease the attack accuracy significantly. It should be noted that the primitive Laplace mechanism frustrates the inference attack badly because the artificial noise added by the mechanism is indiscriminate, leading to conspicuous damages to the model usability. Even so, our solution has significant advantages on MovieLens and MNIST datasets. In contrast, the primitive Laplace mechanism cannot cover image classification cases, while DPSGD cannot defeat the attack.

Table 6: Results of defending the guest against data reconstruction attack.

| $\epsilon$ | MovieLens | | | BookCrossing | | | MNIST | | |
|---|---|---|---|---|---|---|---|---|---|
| | Laplace | DPSGD | **Ours** | Laplace | DPSGD | **Ours** | Laplace | DPSGD | **Ours** |
| 0.1 | 0.2459 | 0.2455 | **0.3223** | 0.3216 | 0.2907 | **0.3329** | 1.8849 | 1.8885 | **2.0181** |
| 0.5 | 0.2453 | 0.2451 | **0.3222** | 0.3202 | 0.2902 | **0.3329** | 1.8024 | 1.8137 | **1.9875** |
| 1.0 | 0.2453 | 0.2451 | **0.3222** | 0.3202 | 0.2902 | **0.3221** | 1.7857 | 1.7509 | **1.9533** |
| 2.0 | 0.2452 | 0.2451 | **0.3222** | 0.3202 | 0.2902 | **0.3221** | 1.7336 | 1.7469 | **1.9391** |
| 4.0 | 0.2452 | 0.2451 | **0.3222** | 0.3202 | 0.2902 | **0.3221** | 1.7014 | 1.7440 | **1.9206** |

***Defense against data reconstruction attack***. We show the defense results against an adversarial host and an adversarial guest in Table 6 and Table 7, respectively. We note that the MSE is measured after the attack model has been trained sufficiently in all cases. The MSEs measured for the attack against baselines of MovieLens, BookCrossing, and MNIST models are 0.2412, 0.2629, and 0.9612 by an adversarial host, 0.2369, 0.2402, and 1.6998 by an adversarial guest, respectively. Please note that these attack results against the baselines are frustrating because the reconstruction attack is hard to succeed in the semi-honest setting. Meanwhile, data samples in two recommendation datasets are similar and embedded with the same feature vectors. This leads to similar reconstruction results and similar MSEs. But we can still conclude from the results that our solution has dominant performance in the defense against reconstruction attacks mounted by either side.

Table 7: Results of defending the host against data reconstruction attack.

| $\epsilon$ | MovieLens | | | BookCrossing | | | MNIST | | |
|---|---|---|---|---|---|---|---|---|---|
| | Laplace | DPSGD | **Ours** | Laplace | DPSGD | **Ours** | Laplace | DPSGD | **Ours** |
| 0.1 | 0.4032 | 0.2417 | **0.5486** | 0.4237 | 0.2758 | **0.5066** | 1.2887 | 1.0875 | **1.8257** |
| 0.5 | 0.4024 | 0.2419 | **0.5357** | 0.4222 | 0.2756 | **0.5149** | 1.2778 | 1.0685 | **1.7758** |
| 1.0 | 0.4008 | 0.2422 | **0.5285** | 0.4217 | 0.2743 | **0.5235** | 1.2602 | 1.0422 | **1.7528** |
| 2.0 | 0.3982 | 0.2421 | **0.5083** | 0.4214 | 0.2697 | **0.5150** | 1.2613 | 1.0333 | **1.7334** |
| 4.0 | 0.3960 | 0.2422 | **0.4819** | 0.4194 | 0.2683 | **0.5046** | 1.2549 | 0.9996 | **1.7262** |

*Defense against feature space hijacking attacks (FSHA)*. Please note that property inference and data reconstruction attacks implemented in [29] hijack the learning objective, offering the adversary an advantage over the previous attacks we straightforwardly adapted from FL. In this setting, the malicious attacker trains a generator using the split neural network as a discriminator during the split learning process and uses a gradient-scaling trick to train the generator. Since the generating part is critical to FSHA, we will focus on the defense against the reconstruction. If the generating part fails, the inference attack will be impossible. Here we give the defense results of the MNIST model because FSHA [29] is mainly evaluated using this dataset. We also evaluate our solution for other datasets against FSHA, and the results are given in the appendix. In Figure 1 and Figure 2, we give the reconstruction results of FHSA mounted by an adversarial host and guest against target samples used in [29]. The second row of two figures shows the results of FSHA against baselines. The following rows show that our solution can effectively preserve private data for both the guest and the host, even the privacy budget is relaxed to 4. More practical privacy budget values for defending against FSHA are presented in the appendix.

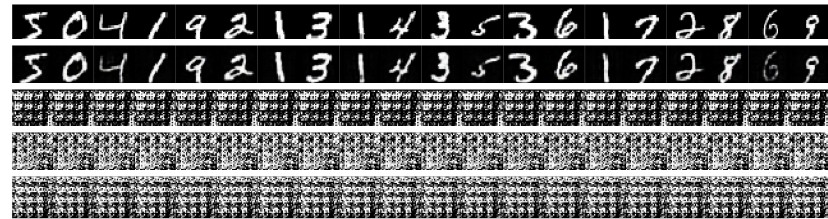

Figure 1: Reconstruction results of FSHA against the guest's data in the first row. The following rows are attack results against the original SplitNN and our solution ($\epsilon = 0.1, 1.0, 4.0$), respectively.

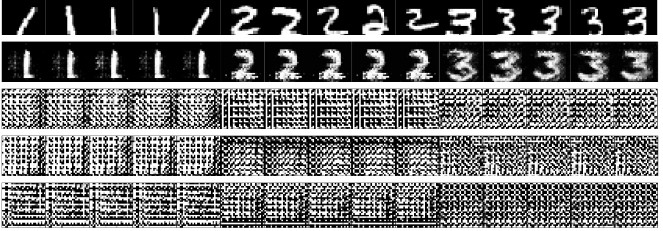

Figure 2: Reconstruction results of FSHA against the host's data in the first row. The following rows are attack results against the original SplitNN and our solution ($\epsilon = 0.1, 1.0, 4.0$), respectively.

## 5 Conclusion and Limitation

Our privacy-preserving SplitNN solution, built upon a new activation function $R^3$eLU and its derivative $R^3$eLU-Diff in a randomized-response manner, significantly reduces privacy leakage risk for both the guest and the host. We show that our solution can provide a tight privacy budget for split learning through the privacy analysis. The model usability and privacy loss can be further balanced by our dynamic privacy budget allocation. The experimental evaluation using different datasets shows that our solution outperforms the existing privacy-preserving SplitNN solutions in model usability and privacy protection. We note that our solution deals with property inference and data reconstruction attacks in a feature level, but a clustering-based label inference attack [8] is out of our reach, which is an interesting topic to be studied in future work.

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

# A  Appendix

## A.1  Split Leaning with R³eLU

We give the forward propagation procedure of SplitNN using our R³eLU in Algorithm 1, and the backward propagation procedure of SplitNN using our R³eLU-Diff in Algorithm 2. The two algorithms are supplementary materials for Section 3.2 and 3.3.

---

**Algorithm 1:** forward propagation with R³eLU

---

**Input:** batch size $N$, data batch $\boldsymbol{x}^g, \boldsymbol{x}^h$, encoded label $\boldsymbol{y}$, training indicator $I$, noise scale $\sigma$.
**Output:** loss or prediction.

1  $\boldsymbol{\theta}^h \xleftarrow{r} \mathcal{N}(0,1)$, $\boldsymbol{\theta}^k \xleftarrow{r} \mathcal{N}(0,1)$, $\boldsymbol{\theta}^g \xleftarrow{r} \mathcal{N}(0,1)$          // initial

  **Guest:**
2  **for** $i \leftarrow 1$ **to** $N$ **do**
3    $\big|$   $v_i^g \leftarrow Embedding(x_i^g)$;
4  **end**
5  $\boldsymbol{v}^g \leftarrow \{v_1^g, v_2^g, \ldots, v_N^g\}$
6  $\boldsymbol{a}^g \leftarrow Forward\text{-}R^3eLU(\boldsymbol{v}^g, \boldsymbol{\theta}^g)$          // send $\boldsymbol{a}^g$ to the host

  **Host:**
7  **for** $i \leftarrow 1$ **to** $N$ **do**
8    $\big|$   $v_i^h \leftarrow Embedding(x_i^h)$;
9  **end**
10  $\boldsymbol{v}^h \leftarrow \{v_1^h, v_2^h, \ldots, v_N^h\}$
11  $\boldsymbol{a}^h \leftarrow Forward(\boldsymbol{v}^h, \boldsymbol{\theta}^h)$          // wait for $\boldsymbol{a}^g$ from the guest
12  $\bar{\boldsymbol{a}} \leftarrow Average(\boldsymbol{a}^g, \boldsymbol{a}^h)$
13  $\boldsymbol{o}^k \leftarrow Forward(\bar{\boldsymbol{a}}, \boldsymbol{\theta}^k)$
14  **if** $I == train$ **then**
15    $\big|$   $loss \leftarrow \mathcal{L}(\boldsymbol{o}^k, \boldsymbol{y})$          // training
16  **else**
17    $\big|$   $pred \leftarrow Softmax(\boldsymbol{o}^k)$          // predicting
18  **end**

---

**Algorithm 2:** backward propagation with R³eLU-Diff

---

**Input:** batch size $N$, feature vectors $\boldsymbol{v}^g, \boldsymbol{v}^h$, forward results $\boldsymbol{a}^g, \boldsymbol{a}^h, \bar{\boldsymbol{a}}$, total loss $l$, encoded label $\boldsymbol{y}$, learning rate $\eta^g, \eta^h, \eta^k$, clipping bound $C$, noise scale $\sigma$.
**Output:** updated parameters $\boldsymbol{\theta}_{t+1}^g, \boldsymbol{\theta}_{t+1}^h, \boldsymbol{\theta}_{t+1}^k$.

  **Host:**
1  $\boldsymbol{\delta}^k \leftarrow \nabla_{\boldsymbol{a}^k} \mathcal{L}(l, \bar{\boldsymbol{a}}, \boldsymbol{\theta}_t^k)$
2  $\boldsymbol{g}_t^k \leftarrow \texttt{Backward}(\boldsymbol{\delta}^k, \bar{\boldsymbol{a}}, \boldsymbol{\theta}_t^k)$
3  $\boldsymbol{\delta}^h \leftarrow \nabla_{\boldsymbol{a}^h} \mathcal{L}(\boldsymbol{\delta}^k, \boldsymbol{a}^h, \boldsymbol{\theta}_t^h)$, $\boldsymbol{\delta}^g \leftarrow \nabla_{\boldsymbol{a}^g} \mathcal{L}(\boldsymbol{\delta}^k, \boldsymbol{a}^g, \boldsymbol{\theta}_t^k)$
4  $\boldsymbol{g}_t^h \leftarrow \texttt{Backward}(\boldsymbol{\delta}^h, \boldsymbol{v}^h, \boldsymbol{\theta}_t^h)$
5  $\bar{\boldsymbol{g}}_t^k \leftarrow \frac{1}{N} \sum_{i \in [1,N]} g_{i,t}^k$, $\bar{\boldsymbol{g}}_t^h \leftarrow \frac{1}{N} \sum_{i \in [1,N]} g_{i,t}^h$
6  $\boldsymbol{\theta}_{t+1}^k \leftarrow \boldsymbol{\theta}_t^k - \eta^k \bar{\boldsymbol{g}}_t^k$, $\boldsymbol{\theta}_{t+1}^h \leftarrow \boldsymbol{\theta}_t^h - \eta^h \bar{\boldsymbol{g}}_t^h$          // host updates
7  $\hat{\boldsymbol{\delta}}^g \leftarrow \boldsymbol{\delta}^g / \max(1, \frac{||\boldsymbol{\delta}^g||_1}{C})$
8  $\tilde{\boldsymbol{\delta}}^g \leftarrow R^3eLU\text{-}Diff(\hat{\boldsymbol{\delta}}^g, \boldsymbol{a}^g, \boldsymbol{v}^g)$          // send $\tilde{\boldsymbol{\delta}}^g$ to the guest

  **Guest:**
9  $\boldsymbol{g}_t^g \leftarrow \texttt{Backward}(\tilde{\boldsymbol{\delta}}^g, \boldsymbol{a}^g, \boldsymbol{\theta}_t^g)$
10  $\bar{\boldsymbol{g}}_t^g \leftarrow \frac{1}{N} \sum_{i \in [1,N]} g_{i,t}^g$
11  $\boldsymbol{\theta}_{t+1}^g \leftarrow \boldsymbol{\theta}_t^g - \eta^g \bar{\boldsymbol{g}}_t^g$          // guest updates

---

## A.2 Privacy Analysis

495 Here we give the proof of our Corollary 1 in detail for the privacy analysis in Section 3.5.

496 *Proof.* Without loss of generality, we assume an activation result $\boldsymbol{a} = \{a_1 = 0, a_2 = 0, ..., a_o = 0, a_{o+1} > 0, ..., a_{N_s} > 0\}$. Then the difference of activations for two neighboring feature vectors $v, v'$ can be bounded by

$$\frac{P(\boldsymbol{a}|v)}{P(\boldsymbol{a}|v')}$$

$$= \prod_{i=1}^{o} \frac{p + \frac{1-p}{2}\exp(-\frac{|f(x)_i|}{b})}{p + \frac{1-p}{2}\exp(-\frac{|f(v')_i|}{b})} \times \prod_{i=o+1}^{N_s} \frac{\frac{1-p}{2b}\exp(-\frac{|a_i - f(v)_i|}{b})}{\frac{1-p}{2b}\exp(-\frac{|a_i - f(v')_i|}{b})}$$

$$= \prod_{i=1}^{o} \frac{p\exp(\frac{|f(v')_i|}{b}) + \frac{1-p}{2}\exp(\frac{|f(v')_i| - |f(v)_i|}{b})}{p\exp(\frac{|f(v')_i|}{b}) + \frac{1-p}{2}} \times$$

$$\prod_{i=o+1}^{N_s} \exp(\frac{|a_i - f(v')_i| - |a_i - f(v)_i|}{b})$$

$$\leq \prod_{i=1}^{o} \frac{p\exp(\frac{|f(v')_i|}{b}) + \frac{1-p}{2}\exp(\frac{|f(v')_i - f(v)_i|}{b})}{p\exp(\frac{|f(v')_i|}{b}) + \frac{1-p}{2}} \times \prod_{i=o+1}^{N_s} \exp(\frac{|f(v')_i - f(v)_i|}{b})$$

$$\leq \prod_{i=1}^{o} \frac{p\exp(\frac{|f(v')_i|}{b})\exp(\frac{-|f(v)_i|}{b}) + \frac{1-p}{2}\exp(\frac{|f(v')_i - f(v)_i|}{b})}{p\exp(\frac{|f(v')_i|}{b})\exp(\frac{-|f(v)_i|}{b}) + \frac{1-p}{2}} \times$$

$$\prod_{i=o+1}^{N_s} \exp(\frac{|f(v')_i - f(v)_i|}{b})$$

$$\leq \prod_{i=1}^{o} \frac{\frac{1+p}{2}}{p\exp(\frac{|f(v')_i| - |f(v)_i|}{b}) + \frac{1-p}{2}} \times \prod_{i=o+1}^{N_s} \exp(\frac{|f(v')_i - f(v)_i|}{b})$$

499 We assume $|f(v)_i| \leq c_g$ and $b \leq \frac{c_g}{ln2}$, then the RHS

$$\leq \prod_{i=1}^{o} \frac{\frac{1+p}{2}}{p\exp(\frac{-c_g}{b}) + \frac{1-p}{2}} \times \prod_{i=o+1}^{N_s} \exp(\frac{|f(v')_i - f(v)_i|}{b})$$

$$\leq \prod_{i=1}^{o} \frac{\frac{1+p}{2}}{\exp(\frac{-c_g}{b})} \times \prod_{i=o+1}^{N_s} \exp(\frac{|f(v')_i - f(v)_i|}{b})$$

$$\leq (\frac{\frac{1+p}{2}}{\exp(\frac{-c_g}{b})})^o \times \exp(\frac{\Delta f}{b})$$

$$= \exp(\epsilon)$$

500 Thus, we have $N_S(\frac{c_g}{\sigma_g} + \ln(\frac{1+p}{2})) + \frac{\Delta f}{\sigma_g} = \epsilon$. $\qquad\square$

## A.3 Evaluation of Dynamic Parameter Importance Estimation

502 Additionally, we evaluate the effectiveness of our feature importance estimation method and give
503 the result here. We compare the estimating results between parameter importance estimated in the
504 model's finally stable state and parameter importance estimated in our dynamic manner. The result
505 is shown in Figure 3. We can conclude from the result that our dynamic parameter importance
506 estimation approach can achieve desirable effectiveness.

507 We aim to estimate the importance of parameters precisely during privacy-preserving split learning.
508 But it is difficult to get the same result with the estimation in a non-perturbed case. By carefully
509 constructing the dynamic estimation method, we can obtain dynamic estimation results of noisy pa-
510 rameters quite close to a baseline estimation result in the stable state without any privacy protection.
511 In Figure 4, we show the result of our dynamic importance estimation approach. We can find that

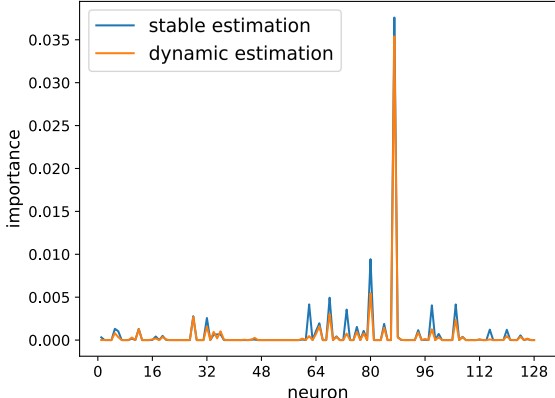

Figure 3: Comparison of parameter importance estimation results.

artificial noise introduced by our privacy-preserving SplitNN solution will affect the estimation of parameter importance. But the growth tendency keep the same as the baseline. This is good enough for us since we use the proportionality factor of each parameter to calculate the budget allocation, which will not be influenced by the magnitude.

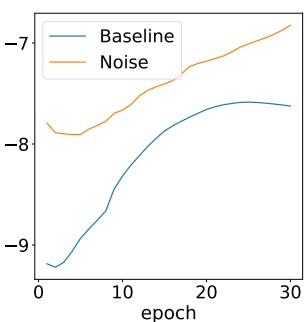 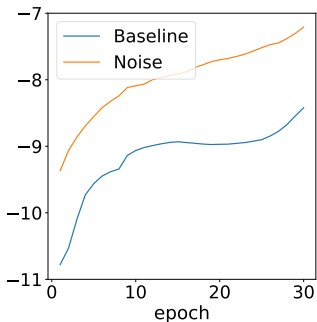

Figure 4: Importance of noisy parameters estimated dynamically in logarithm.

## A.4   Supplementary Results for Evaluation

The neural networks we used for the MovieLens, BookCrossing, and MNIST datasets after split are shown in Tab 8, Table 9, and Table 10. These networks are commonly used in related studies. We split them according to the interpretation of SplitNN in [4, 11, 29].

| Guest Layer | Output Shape | Param # |
|---|---|---|
| Linear(160,128)+ReLU | (None,128) | 20608 |

| Host Layer | Output Shape | Param # |
|---|---|---|
| Linear(160,128)+ReLU | (None,128) | 20608 |
| Merge Guest | | |
| Linear(128,128)+ReLU | (None,128) | 16512 |
| Linear(128,64)+ReLU | (None,64) | 8256 |
| Linear(64,3952)+Softmax | (None,3952) | 256880 |

Table 8: The MovieLens model.

To further investigate how our solution affects the learning process of SplitNN, we report learning results of a MovieLens recommendation model protecting the privacy of the guest and the host in Figure 5 and 6, respectively. In each plot, we show trends of training loss, training accuracy, and testing accuracy as the training epoch increases. We can conclude from these figures that our solution achieve satisfying model usability with small privacy budget for either side of SplitNN.

| Guest Layer | Output Shape | Param # |
|---|---|---|
| Linear(160,128)+ReLU | (None,128) | 20608 |

| Host Layer | Output Shape | Param # |
|---|---|---|
| Linear(160,128)+ReLU | (None,128) | 20608 |
| Merge Guest | | |
| Linear(128,256)+ReLU | (None,256) | 33024 |
| Linear(256,128)+ReLU | (None,128) | 32896 |
| Linear(128,17384)+Softmax | (None,10) | 2242536 |

Table 9: The BookCrossing model.

| Guest Layer | Output Shape | Param # |
|---|---|---|
| Linear(28*14,128) | (None,128) | 50304 |
| BatchNorm+ReLU | (None,128) | |
| Linear(128,64) | (None,64) | 8256 |

| Host Layer | Output Shape | Param # |
|---|---|---|
| Linear(28*14,128) | (None,128) | 50304 |
| BatchNorm+ReLU | (None,128) | |
| Linear(128.64) | (None,64) | 8256 |
| BatchNorm+ReLU | (None,64) | |
| Merge Guest | | |
| Linear(64,64) | (None,64) | 4160 |
| BatchNorm+ReLU | (None,64) | |
| Linear(64,10)+Softmax | (None,10) | 650 |

Table 10: The MNIST model.

We notice that FSHA is sensitive to our privacy-preserving SplitNN solution. In Figure 1 and Figure 2, we find that neither an adversarial host nor an adversarial guest can reconstruct meaningful samples, even if the target's privacy budget is $\epsilon = 4$. Therefore, we are curious about a practical choice of the privacy budget when dealing with FSHA. To this end, we give more defense results of our solution against FSHA using various privacy budget values in Table 11. We can conclude from the MSE results that two recommendation models prefer relatively low privacy budget, such as $\epsilon = 1.0$. However, it is interesting to see that FSHA attack against an image classification model using SplitNN can be frustrated by our solution using relatively high privacy budget, which also means a high model usability.

Table 11: Defense results against FSHA mounted by an adversarial host.

| $\epsilon$ | MovieLens | | BookCrossing | | MNIST | |
|---|---|---|---|---|---|---|
| | baseline | **Ours** | baseline | **Ours** | baseline | **Ours** |
| | 1200 Epochs | 5000 Epochs | 1200 Epochs | 5000 Epochs | 9000 Epochs | 9000 Epochs |
| 0.1 | | **455.5676** | | **500.1487** | | **1.98257** |
| 0.25 | | **73.7824** | | **83.7606** | | **1.9788** |
| 0.5 | | **21.1706** | | **30.2905** | | **1.9703** |
| 0.75 | | **9.5312** | | **8.5984** | | **1.9534** |
| 1.0 | $0.2652 \times 10^{-3}$ | **4.4903** | $0.2365 \times 10^{-3}$ | **6.1267** | 0.0206 | **1.9442** |
| 2.0 | | **1.1559** | | **1.7047** | | **1.9283** |
| 4.0 | | **0.2719** | | **0.2478** | | **1.9209** |
| 6.0 | | **0.1091** | | **0.1638** | | **1.9135** |
| 8.0 | | **0.0975** | | **0.0846** | | **1.9116** |

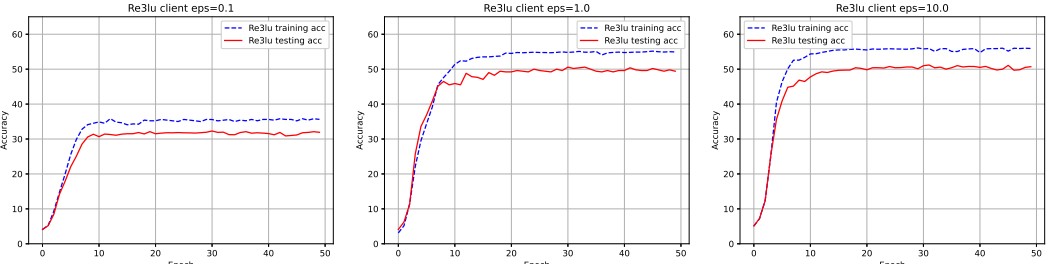

Figure 5: SplitNN learning curve with the guest's privacy protected by our solution.

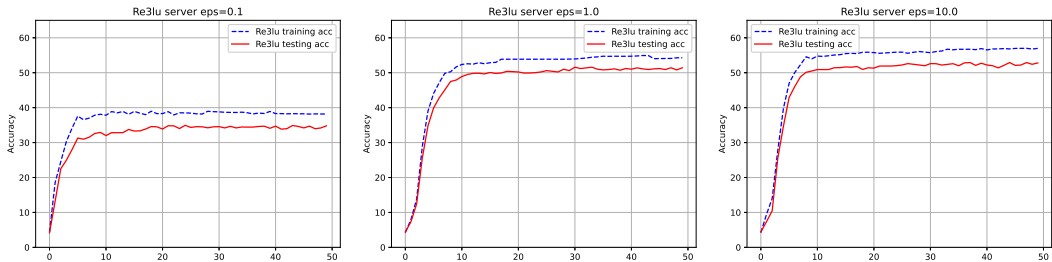

Figure 6: SplitNN learning curve with the host's privacy protected by our solution.

