# OpenReview forum: "Secure Split Learning against Property Inference and Data Reconstruction Attacks"
_NeurIPS.cc/2022/Conference — NeurIPS 2022 Submitted_

### Official Review · Reviewer_QQtJ · 2022-07-09

**Rating:** 4
**Confidence:** 3
**Soundness:** 2 fair
**Presentation:** 2 fair
**Contribution:** 2 fair

**Summary:**

This paper presents a way to improve privacy in split learning by constructing a privacy-tunnel between the host and the guest. This is done by using a new activation function called R3eLU, which responds to forward and backward passes in a randomized manner. The noise added to the activation through this activation function provides differential privacy on the activations and its gradients . The proposal is evaluated on recommendation and classification models to show that it improves the utility-privacy tradeoff of split-NNs.

**Questions:**

1. Privacy Amplification: The analysis presented in the paper assumes that the random sampling in SplitNN provides a privacy amplification of B/N. My understanding is that privacy amplification by subsampling requires the input-ids to be unknown to the host. However, the guest needs to share the input-id of the samples in the batch with the host to coordinate the training. Can you comment on this?

**Limitations:**

The paper discusses the limitations of the proposed defense.

**Strengths And Weaknesses:**

## Strength

1. The idea of using a randomized activation function to add noise during forward and backward propagation to provide DP is interesting.
2. The idea seems to provide better protection against property inference and reconstruction attacks compared to other forms of noise addition (DPSGD and Laplace)



## Weakness

1. The writing in the evaluation section is unclear and is missing key details about the experimental setup.
2. Property inference: It’s unclear what this means in the context of the paper. Is the adversary trying to infer additional attributes from the data? If so, what are these additional attributes for the datasets used in the results?
3. How is DPSGD being applied to split learning?  Is the noise being applied to the embedding or the gradients shared during backpropagation or both?

---

> ### Author Response · Authors · 2022-08-02
> **Response to reviewer QQtJ**
>
> W1. [unclear description] Thanks for your advice. We will modify the evaluation section and supply more details of the experiments in the revision.
>
> W2. [property inference attack] A property inference attack is to infer an existing property (or attribute) of the data samples. For example, an adversarial host in our experiments infers the age attribute of the guest's data. However, the host has no idea of the age since training data is vertically partitioned.
>
> W3. [how is DP-SGD applied] We apply DP-SGD in the same way as the original paper, i.e., applied to gradients. In addition to the comparison, we add Laplace noises to embeddings in the forward passing when protecting the guest's private data.
>
> Q1. The reviewer's intuition is right. In split learning, or more generally, in vertically partitioned collaborative learning, the participants need to align the training dataset first. This is usually done by running a private set intersection (PSI) protocol and finding participants' common IDs of training data. Therefore, input data IDs of each batch should be common knowledge for combining attributes correctly.

---

### Official Review · Reviewer_Yh7T · 2022-07-11

**Rating:** 6
**Confidence:** 4
**Soundness:** 2 fair
**Presentation:** 2 fair
**Contribution:** 2 fair

**Summary:**

The authors present a differential privacy mech for training with split learning. Their method is based on a randomized response like mech applied to privatizing relu activations.

**Questions:**


-Typically the privacy budgeting is about composition laws, or dealing with sampling with (repitition) replacement across batches for the same.
- Why is your proposed budgeting based on perceived feature importance instead?


**Limitations:**

Please answer the concerns under questions and Weaknesses. That will help clarify whether those fall under limitations or not.

**Strengths And Weaknesses:**

Strengths:

-Earliest of DP mechs for split learning
-

Weakness:

-Typically the privacy budgeting is about composition laws, or dealing with sampling with (repitition) replacement across batches for the same.
- Why is your proposed budgeting based on perceived feature importance instead?

- For DP-SGD, typically the model weights as a query are a function of all of training data. As opposed to that in terms of making activations private, the query is 1 sample at a time. Each activation is a function of 1 sample. This brings it under a local dp like regime.
- This needs to be stated along with reasoning of why the proposed method is comparable with DP-SGD in terms of privacy models or paradigms.

Minor/typo: Change Split Leaning to Split Learning in A.1

---

> ### Author Response · Authors · 2022-08-02
> **Response to reviewer Yh7T**
>
> Thanks for the comments and questions. We agree with the reviewer that privacy budgeting in DP nowadays mainly focuses on composition and subsampling ways. Related work has made remarkable achievements in various mechanism designs. We believe composition with fine-grained analysis (indeed what we have done in the analysis) and subsampling data samples are well-established methods of practical DP mechanisms. However, we think these should not be the only ways. When calculating privacy loss for a DP mechanism, we should start with the queried data and then query functions until the output. Informally, we can say that subsampling is an approach to tightening the privacy budget by manipulating the data to be queried, while composition research is the precise accountant for the privacy loss of the output. We, in this paper, try to construct a new query function with the proposed R3eLU, which is orthogonal to subsampling and composition studies. However, we can use these techniques to improve our mechanism further.
>
> The feature importance is used for dynamic privacy budget allocation. We can go through our privacy-preserving solution works in this way. First, we replace activation functions in the neural network of a split learning task with our R3eLU function. Then, we estimate the importance of each feature representation in the network. Next, we use the feature importance to weight a unit of privacy budget and perturb the activating outputs using the corresponding randomness. Finally, we sum up all privacy budgets for each feature representation as the total privacy budget. However, we can also do privacy budgeting in another way (similar to DP-SGD). The first two steps are the same as the previous. In the third step, we allocate a total privacy budget to each feature representation according to the estimated importance and stop the training when we run out of the budget.
>
> The main difference between DP-SGD and our solution is that we perturb the output of a query of training data by fuzzing the query function in a randomized response way, while DP-SGD fuzzes the query function by perturbing the coefficients of the function. Since the adversary of membership (or property) inference attacks against split learning leverages the intermediate result (activations of the guest, partial loss of the host) for attacking, it is better to defend attacks at the root, i.e., preserving activations of the guest (with R3eLU) and partial losses of the host (with R3eLU-Diff). In this way, we can achieve a better defensive performance than DP-SGD. As for the privacy comparison, we choose DP-SGD because there are no DP mechanisms for split learning to be compared, and DP-SGD is a mature technique for the general learning paradigm.
>
> Additionally, our work is dedicated to preserving the data privacy of the guest and the host in split learning. The privacy model is slightly different from DP-SGD. We make activations and partial losses private since the attacks against split learning depend on them.

---

> > ### Comment · Reviewer_Yh7T · 2022-08-07
> > **Heuristic or proof?**
> >
> > Is the notion of assigning budget based on estimated feature importance, a heuristic to privacy budgeting or is it justified by a proof? Can authors point out to that proof of justification?

---

> > > ### Author Response · Authors · 2022-08-08
> > > **Heuristic**
> > >
> > > Thanks for your concern. Assigning the privacy budget based on estimated feature importance in split learning is yet heuristic. This notion aims to improve the model performance given a fixed privacy budget. We give a primary design of dynamic budget allocation in Equation (6) in the paper. The feasibility of our method is proved by experimental results (shown in Appendix A.3). We note that a formal analysis and proof of privacy budgeting is of particular interest but tricky and should be thoroughly studied in a separate work. If a strong result could be obtained, many relevant studies will benefit, including our work. It should also be noted that privacy budget allocation in deep learning is forming a hot topic like other resource allocation problems. Some privacy budgeting studies from different aspects (iteration [r1], scheduling in FL [r2]) may also interest the reviewer. We remark that our privacy-preserving split learning with R3eLU can work in a standard privacy budgeting strategy or an optimized one. In other words, the proposed R3eLU is orthogonal to privacy budgeting studies and our solution can be further improved by more effective budgeting strategies.
> > >
> > > [r1] Lee, Jaewoo, and Daniel Kifer. "Concentrated differentially private gradient descent with adaptive per-iteration privacy budget." Proceedings of the 24th ACM SIGKDD International Conference on Knowledge Discovery & Data Mining. 2018.
> > > [r2] Luo, Tao, et al. "Privacy budget scheduling." 15th {USENIX} Symposium on Operating Systems Design and Implementation ({OSDI} 21). 2021.

---

> > > > ### Comment · Reviewer_Yh7T · 2022-08-08
> > > > **reply**
> > > >
> > > > if your privacy budgeting is via a heuristic and not a proof, how can you ensure an eps-dp for the end user? User requests eps-dp and maybe you have > eps that is being provided in reality. isn't this a major issue?

---

> > > > > ### Author Response · Authors · 2022-08-09
> > > > > **Heuristic feature importance estimation for budget allocation**
> > > > >
> > > > > Thanks for your reply. First of all, we apologize that our inappropriate explanations caused misunderstanding. We realized there was a misunderstanding during the discussion. **The privacy leakage caused will not exceed the privacy budget in our solution, which has been formally proved in the privacy analysis part**. We misunderstood the reviewer's intention regarding the question of "assigning budget based on estimated feature importance." The **heuristic part**, as we explained in the previous post, is feature importance estimation (for privacy budget allocation). Since it is hard for an end-user to determine the privacy budget for each feature, we allow the end-user to give a total privacy budget. Once determined, our solution will not exceed it during the allocation.
> > > > >
> > > > > Determining the privacy budget precisely for each feature (privacy budget allocation) is complex. We use feature importance estimation as a heuristic method in the paper. That is the main point of our previous post.
> > > > >
> > > > > We do privacy budgeting at a feature level, which means private information leaked in each feature will be controlled by the budgeting. For example, if a privacy budget of a specific feature is zero, then the activating results relevant to the feature will be eliminated, leaking no information to the following computation. Once an end-user determines the total privacy budget (epsilon), our solution will allocate the budget to each feature equally (in the standard mode) or dynamically (in the feature importance estimation mode), say epsilon_1,epsilon_2, ..., epsilon_N. By combing these N feature-level DP mechanisms, we get an epsilon-DP mechanism for the end-user. Therefore, the total leakage will not exceed the user's budget since epsilon_1,epsilon_2, ..., epsilon_N will be constrained by our solution. This has been proved in the paper. Please refer to Corollary 1, Corollary 2 on page 6, and the analysis in Appendix A.2. We hope this reply can resolve the misunderstanding, and we are glad to have more discussions with the reviewer.

---

> > > > > > ### Comment · Reviewer_Yh7T · 2022-08-09
> > > > > > **Satisfactory**
> > > > > >
> > > > > > Thank you for the clarification.

---

### Official Review · Reviewer_Nq3T · 2022-07-11

**Rating:** 5
**Confidence:** 3
**Soundness:** 3 good
**Presentation:** 3 good
**Contribution:** 2 fair

**Summary:**

The paper investigates adversarial effects of property inference and data reconstruction attacks in the setting of split learning for features partitioned vertically. To defend against such attacks, the paper proposes a privacy-preserving technique that involves adding noise in the ReLU activation during the forward and backward propagation steps in the split learning protocol. In addition, it proposes a dynamic allocation of the privacy budget through estimating the importance of features.


**Questions:**

Q1. Why is the influence estimation of single features considered and not the combination of features? I.e., could it be that the single feature change might not have an influence on the output but that a combination of features is particularly vulnerable? In this case, why did you consider single feature influence estimation?

Q2. Why is DPSGD much worse at defending against property inference attacks than the Laplace or your proposed solution (e.g., in Table 4 and Table 5 there are almost 20% differences in prediction accuracy for BookCrossing and more than 30% for MNIST)?

Q3. Does it make sense to evaluate the data reconstruction attack using MSE? At least for MNIST, should we be looking at measures of visual similarity, e.g., Frechet distance?

Q4. What is the baseline mechanism used for Laplace? Does it refer to applying the perturbation without the fine-grained analysis? It is not mentioned in the evaluation.

Q5. Why does the property inference attack by the adversarial host have a much higher 94% accuracy on MNIST compared to the 57% accuracy by the adversarial guest?

Q6. What do you mean by “that these attack results against the baselines are frustrating because the reconstruction attack is hard..” or “ … primitive Laplace mechanism frustrates the inference attack…” (perhaps deteriorates/inhibits the attack accuracy?)?

**Strengths And Weaknesses:**

The paper tackles a new topic in split learning. Overall, I found the paper clear and significant, though there were some unnatural phrasings here and there and some typos. The paper offers an original approach to privacy-preserving split learning but the evaluation could be more insightful and explain a bit more certain phenomena (see Questions).

**Strengths**

S1. The proposed technique is privacy-preserving, based on the formal guarantee of differential privacy.

S2. Interesting technique to apply the perturbation at the activation level and propose $R^3eLU$.

S3. The paper argues in the evaluation section from the perspective of better defenses against adversarial attacks, which both improve with the Laplace mechanism.

**Weaknesses**

W1. The evaluation results are not that impressive in terms of the model utility (the utility is up to 2% better compared to DPSGD), though they are better with the fine-grained analysis (Ours in Table 2 and 3). It seems that for these datasets even DPSGD comes pretty close to the baseline training (no privacy) of SplitNN.

W2. Not clear what the overhead (in performance) is for the dynamic budget allocation (if any) over baseline (no privacy).

---

> ### Author Response · Authors · 2022-08-02
> **Response to reviewer Nq3T**
>
> W1. [evaluation result not impressive] We note that the model performance of our solution is close to the baseline (without any privacy protection) performance given in Table 1 in the paper. However, the model performance of DP-SGD is relatively low, as indicated in Table 2 in the paper. We can find that our solution can achieve better performance in most cases compared with naive Laplace and DP-SGD. DP-SGD has a close model accuracy to our solution because the perturbation used in DP-SGD will be mitigated during splitting learning. Since the guest and the host apply DP-SGD to their gradients individually, the perturbation of either side cannot provide a consistent influence on the whole model.
>
> W2. [overhead of budget allocation] For the dynamic budget allocation, after each epoch, we fetch the gradients of parameters and use the one-order Taylor expression to approximate the importance, which is the multiplication of the gradient and the weight. As for the importance of a feature, we calculate the quadratic sum of the weights which are relevant to the feature. The most important operations during the calculation are matrix multiplications and summations in the scale of model parameters. Therefore, our solution will not cause much computational overhead compared to the baseline.
>
> Q1. This is due to three main reasons. On one side, our purpose is to protect the membership and features of training data. It is more effective to stem the leakage from the root. On the other side, a DP mechanism gives a post-processing property, which ensures that the composition of several DP outputs of querying features is still privacy-preserving. Moreover, the paper uses a fine-grained influence estimation that only considers the influence of single features. As a result, the computational overhead is much lower than estimating the influence of all possible combinations. Overall, it is reasonable to compute single feature importance rather than the combinations.
>
> Q2: DPSGD is designed for general neural network learning paradigms, which will disturb only the gradients. However, the Laplace mechanism and our proposed solution directly disturb the outputs, including the intermediate results (activations in the forward process and partial loss in the backward process). We believe that the perturbation in the outputs will also cause the perturbation in the gradients. However, the perturbation on gradients will be mitigated in the forward and backward passes before it can be applied to the attack surface. In brief, our solution and the Laplace noise are directly applied to the attack surface, having a much shorter decay path than DP-SGD. That is why DP-SGD does not perform well in defending against property inference attacks. It should also be noted that privacy preservation may take model performance as a cost. DP-SGD gains more model performance when it has a longer perturbation decay path. This can be considered together with the reviewer's first concern listed in the Weakness.
>
> Q3: We agree with the reviewer that more proper metrics should be taken into account for the evaluation. As far as we know, MSE is still the most prevalent metric in data reconstruction evaluation. Since we propose the solution to defending against some known studies, it is straightforward to use the same metric as attack studies like property inference and FSHA [29].
>
> Q4: We use the Laplace mechanism as a comparison. We clip the intermediate results for bounding the sensitivity as we do in our proposed solution. Then we directly draw the noise from the corresponding Laplace distribution and add it to the intermediate results.
>
> Q5: We have noticed this phenomenon and found it reasonable. The key reason may be that the host can acquire more raw information than the guest if we partition the attributes equally. This is due to the network architecture of split learning. The smashed data of the client is available to the host, but the smashed data of the host is not available to the guest. Only the corresponding partial loss may leak limited information about the host to the guest. Therefore, the adversarial host could have better attack performance than an adversarial guest. The same phenomenon has also been reported in recent attack studies like FSHA [29].
>
> Q6: Yes, these sentences express the same meaning as "deteriorates/inhibits the attack accuracy". As indicated by the reviewer, the "frustrate" here means "deteriorate". By the first sentence, we mean that a reconstruction attack is hard to have a significant effect in this semi-honest setting. By the second sentence, we mean that the Laplace noise will badly deteriorate the inference attack since the noise is indiscriminate.

---

> > ### Comment · Reviewer_Yh7T · 2022-08-07
> > **Model perf is comparable with baseline**
> >
> > This resolves the model performance comparison issue raised by that reviewer.

---

### Meta-Review · Area_Chair_kbe7 · 2022-08-27

**Recommendation:** Reject
**Confidence:** Certain

**Metareview:**

Though reviewers increased their scores, they maintained some skepticism regarding several issues in the paper. The authors are strongly encouraged to consider addressing these in a later submission.

- Privacy amplification by subsampling seems to require that specific inputs used in a batch (after subsampling) should not be known to the host. This condition does not seem to be readily satisfied in split-learning as the input-ids need to be communicated to the host during training.
- Gains remain modest compared to simple baselines.
- Steps within the pipeline remain heuristics, and are not formally justified.

**Award:**

No

---

### Decision · Program_Chairs · 2022-09-14

Reject